# Horizon: Facebook's Open Source Applied Reinforcement Learning Platform

**Jason Gauci** [1]   **Edoardo Conti** [1]   **Yitao Liang** [1]   **Kittipat Virochsiri** [1]   **Yuchen He** [1]   **Zachary Kaden** [1]
**Vivek Narayanan** [1]   **Xiaohui Ye** [1]   **Zhengxing Chen** [1]

## Abstract

In this paper we present Horizon, Facebook's open source applied reinforcement learning (RL) platform. Horizon is an end-to-end platform designed to solve industry applied RL problems where datasets are large (millions to billions of observations), the feedback loop is slow (vs. a simulator), and experiments must be done with care because they don't run in a simulator. Unlike other RL platforms, which are often designed for fast prototyping and experimentation, Horizon is designed with production use cases as top of mind. The platform contains workflows to train popular deep RL algorithms and includes data preprocessing, feature transformation, distributed training, counterfactual policy evaluation, optimized serving, and a model-based data understanding tool. We also showcase and describe real examples where reinforcement learning models trained with Horizon significantly outperformed and replaced supervised learning systems at Facebook.

## 1. Introduction

Deep reinforcement learning (RL) is poised to revolutionize how autonomous systems are built. In recent years, it has been shown to achieve state-of-the-art performance on a wide variety of complicated tasks (Mnih et al., 2015; Lillicrap et al., 2015; Schulman et al., 2015; Van Hasselt et al., 2016; Schulman et al., 2017), where being successful requires learning complex relationships between high dimensional state spaces, actions, and long term rewards. However, the current implementations of the latest advances in this field have mainly been tailored to academia, focusing on fast prototyping and evaluating performance on simulated benchmark environments.

---

[1]Facebook, Menlo Park, California, USA. Correspondence to: Jason Gauci <jjg@fb.com>, Edoardo Conti <edoardoc@fb.com>.

*Reinforcement Learning for Real Life (RL4RealLife) Workshop in the 36th International Conference on Machine Learning*, Long Beach, California, USA, 2019. Copyright 2019 by the author(s).

*Table 1.* **Comparison of Open Source RL Frameworks.** DP = Data Preprocessing & Feature Normalization, DT = Distributed Training, CPE = Counterfactual Policy Evaluation, EC2 = Amazon EC2 Integration.

| FRAMEWORK | DP | DT | CPE | EC2 |
|---|---|---|---|---|
| HORIZON | ✓ | ✓ | ✓ | × |
| GARAGE | × | ✓ | × | ✓ |
| DOPAMINE | × | × | × | × |
| COACH | × | ✓ | × | × |
| SAGEMAKER RL | × | ✓ | × | ✓ |

While interest in applying RL to real problems in industry is high (Chen et al., 2019; Zhao et al., 2018b;a; Mirhoseini et al., 2017; Zheng et al., 2018), the current set of implementations and tooling must be adapted to handle the unique challenges faced in applied settings. Specifically, the handling of large datasets with hundreds or thousands of varying feature types and distributions, high dimensional discrete and continuous action spaces, optimized training and serving, and algorithm performance estimates before deployment are of key importance.

Currently, several platforms have been developed that address different parts of this end-to-end applied RL challenge (Bellemare et al., 2018; Caspi et al., 2017; Liang et al., 2017; Agarwal et al., 2016), however to our knowledge, no single system offers an end-to-end solution. Table 1 outlines the features of different frameworks compared to Horizon.

With this in mind, we introduce Horizon - an open source end-to-end platform for applied RL developed and used at Facebook. Horizon is built in Python and uses PyTorch for modeling and training (Paszke et al., 2017) and Caffe2 for model serving (Jia et al., 2014). It aims to fill the rapidly-growing need for RL systems that are tailored to work on real, industry produced, datasets.

The rest of this paper goes into the details and features of Horizon, but at a high level Horizon features:

**Data preprocessing:** A Spark (Zaharia et al., 2010) pipeline that converts logged training data into the format required for training numerous different deep RL models.

**Feature Normalization:** Logic to extract metadata about every feature including type (float, int, enum, probability, etc.) and method to normalize the feature. This metadata is then used to automatically preprocess features during training and serving, mitigating issues from varying feature scales and distributions which has shown to improve model performance and convergence (Ioffe & Szegedy, 2015).

**Data Understanding Tool:** RL algorithms are suitable for sequential problems where some form of accumulated rewards are to be optimized. In contrary to many academic research environments that have well-defined transition and reward functions (Brockman et al., 2016), real world environments are not easily formulated to the standard Markov Decision Process (MDP) framework (Bellman, 1957) with properly defined states, actions, rewards, and transitions. Thus, we developed a data understanding tool that checks properties of problem formulation prior to applying any RL algorithm. In practice, the data understanding tool has accelerated data engineering iterations and provided explainable insights to RL practitioners.

**Deep RL model implementations:** Horizon provides implementations of Deep Q-networks (DQN) (Mnih et al., 2015), Deep Q-networks with double Q-learning (DDQN) (Van Hasselt et al., 2016), Deep Q-networks with dueling architecture (Dueling DQN & Dueling DDQN) (Wang et al., 2015) for discrete action spaces, a parametric action version of all the previously mentioned algorithms for handling very large discrete action spaces, and Deep Deterministic Policy Gradients (DDPG) (Lillicrap et al., 2015) and Soft Actor-Critic (SAC) (Haarnoja et al., 2018) for continuous action spaces.

**Multi-Node and Multi-GPU training:** Industry datasets can be very large. At Facebook many of our datasets contain tens of millions of samples per day. Horizon has functionality to conduct training on many GPUs distributed over numerous machines. This allows for fast model iteration and high utilization of industry sized clusters. Even for problems with very high dimensional feature sets (hundreds or thousands of features) and millions of training examples, we are able to learn models in a few hours (while doing preprocessing and counterfactual policy evaluation on every batch). Horizon supports CPU, GPU, multi-GPU, and multi-node training.

**Counterfactual policy evaluation:** Unlike in pure research settings where simulators offer safe ways to test models and time to collect new samples is very short, in applied settings it is usually rare to have access to a simulator. This makes offline model evaluation important as new models affect the real world and time to collect new observations and re-train models may be days or weeks. Horizon scores trained models offline using several well known counterfactual policy evaluation (CPE) methods. The step-wise importance sampling estimator, step-wise direct sampling estimator, step-wise doubly-robust estimator (Dudík et al., 2011), sequential doubly-robust estimator (Jiang & Li, 2016)[1], and MAGIC estimator (Thomas & Brunskill, 2016) are all run as part of Horizon's end-to-end training workflow.

**Optimized Serving:** Post training, models are exported from PyTorch to a Caffe2 network and set of parameters via ONNX (Exchange, 2018). Caffe2 is optimized for performance and portability, allowing models to be deployed to thousands of machines.

**Tested Algorithms:** Testing production RL systems is a new area with no established best practices. We take inspiration from systems best practices and test our algorithms in Horizon via unit tests and integration tests. Using custom environments (i.e. Gridworld) and some standard environments from OpenAI's Gym (Brockman et al., 2016) we train and evaluate all of our RL models on every pull request.

We end the paper discussing examples of how models trained with Horizon outperformed supervised learning and heuristic based policies to send notifications and to stream videos at Facebook. We provide details into the formulation and methods used in our approach to give practitioners insight into how to successfully apply RL to their problems.

## 2. Data Preprocessing

Many RL models are trained on consecutive pairs of state/action tuples (DQN, DDPG, SAC etc.). However, in production systems data is often logged as it comes in, requiring offline logic to join the data in a format suitable for RL. To assist in creating data in this format, Horizon includes a Spark pipeline (called the *Timeline* pipeline) that transforms logged data collected in the following row format:

- *MDP ID*: A unique ID for the Markov Decision Process (MDP) chain that this training example is a part of.

- *Sequence Number*: A number representing the location of the state in the MDP (i.e. a timestamp).

- *State Features*: The features of the current step that are independent of the action.

- *Action*: The action taken at the current step. A string (i.e. 'up') if the action is discrete or a set of features if the action is parametric or continuous.

- *Action Probability*: The probability that the current system took the action logged. Used in counterfactual policy evaluation.

---

[1]Two variants are implemented; one makes uses of ordinal importance sampling and the other weighted importance sampling.

- *Metrics*: A map from metric name to value. Used to construct a reward value during training by computing the dot product between input weights and metric values.

- *Possible Actions*: An array of possible actions at the current step, including the action chosen (left blank for continuous action domains). This is optional but enables Q-Learning (vs. SARSA).

This data is transformed into data in the row format below. Note, *MDP ID*, *Sequence Number*, *State Features*, *Action*, *Action Probability*, and *Metrics* are also present in the data below, but are left out for brevity.

- *Next State Features*: The features of the subsequent step that are action-independent.

- *Next Action*: The action taken at the next step.

- *Sequence Number Ordinal*: A number representing the location of the state in the MDP after the *Sequence Number* was converted to an ordinal number.

- *Time Diff*: A number representing the "time difference" between the current state and next state (computed as the difference in non-ordinal sequence numbers between states). Used as an optional way to set varying time differences between states. Particularly useful for MDPs that have been sub-sampled upstream.

- *Possible Next Actions*: A list of actions that were possible at the next step. Only present if *Possible Actions* were provided.

As seen above, instead of taking in a reward scalar explicitly, Horizon takes in a "metrics" map. This enables reward shaping during training and counterfactual policy evaluation over metrics.

1. *Reward shaping*: By taking the dot product between the vector of values in the metrics map and the vector of weights in a "metrics weight" map provided by the user at training time, we compute the reward scalar value for each training observation. This allows for rapid iteration on reward shaping. The user can experiment with different reward formulas by specifying different input weights as the input to the training process without the need to regenerate data tables.

2. *Counterfactual policy evaluation over metrics*: The metrics map also enables Horizon's counterfactual policy evaluation pipeline to run over each metric in the map instead of just aggregate reward. This allows for a granular estimation on the newly trained policy's performance.

## 3. Feature Normalization

Data from recommender systems is often sparse, noisy and arbitrarily distributed (Adomavicius & Tuzhilin, 2005). Literature has shown that neural networks learn faster and better when operating on batches of features that are normally distributed (Ioffe & Szegedy, 2015). In RL, where the recurrence can become unstable when exposed to very large features, feature normalization is even more important. For this reason, Horizon includes a workflow that automatically analyzes the training dataset and determines the best transformation function and corresponding normalization parameters for each feature. Developers can override the estimation if they have prior knowledge of the feature that they prefer to use.

In the workflow, features are identified to be of type binary, probability, continuous, enum, quantile, or boxcox. A "normalization specification" is then created which describes how the feature should be normalized during training.

Although we pre-compute the feature transformation functions prior to training, we do not apply the feature transformation to the dataset until during training. At training time we create a PyTorch network that takes in the raw features and applies the normalization during the forward pass. This allows developers to quickly iterate on the feature transformation without regenerating the dataset. The feature transformation process begins by grouping features according to their identity and then processing each group as a single batch using vector operations.

## 4. Data Understanding Tool

One big challenge of applied RL is problem formulation. RL algorithms are theoretically designed on the Markov Decision Process (MDP) framework (Bellman, 1957) where some sort of long-term reward is optimized in a sequential setting. MDP tasks are defined by $(\mathcal{S}, \mathcal{A}, T, R)$ tuples where $\mathcal{S}$ and $\mathcal{A}$ refer to the state and action spaces; $T : \mathcal{S} \times \mathcal{A} \rightarrow \mathcal{S}$ refers to the state transition function, which can be stochastic; and $R : \mathcal{S} \times \mathcal{A} \rightarrow \mathbb{R}$ represents the reward function which maps a transition into a real value. Since this formulation can be unfamiliar to engineers inexperienced in RL, it is easy to accidentally prepare data that does not conform well to the MDP definition. Applying RL on ill-formulated problems is a costly process: (1) online testing RL models trained on wrongly defined environments can regress online metrics; (2) engineering time may be spent debugging and tuning the RL model training process for irrelevant factors such as hyper-parameters.

In order to quickly pre-screen the problem formulation and accelerate feature engineering iterations, we developed a data understanding tool. Using a data-driven, model-based method together with heuristics, it checks whether several

important properties of the problem formulation conform to the MDP framework.

First, the tool learns a model about the formulated environment based on the same dataset to be used in RL training. While there have been extensive research in model-based RL (Deisenroth & Rasmussen, 2011; Nagabandi et al., 2018; Finn & Levine, 2017; Watter et al., 2015) in the line of modeling environments, we use a probabilistic generative model that is capable of handling high-dimensional input and stochasticity of state transitions and rewards, inspired by recent model-based work (Ha & Schmidhuber, 2018). The chosen model is a deep neural network with the input as the current state and action. To handle possible stochasticity in rewards and transitions, the last layer of the neural network is set as a Gaussian Mixture Model (GMM) layer (Bishop, 1994; Variani et al., 2015) such that the model outputs a Gaussian mixture distribution of next states and rewards rather than point estimates:

$$P(s_{t+1}|s_t, a_t) = \sum_k \pi_k \mathcal{N}(\mu_k, \Sigma_k) \qquad (1)$$

We omit the expression of $P(r_t|s_t, a_t)$ since it has a similar form. In Eqn. 1, $k$ is a hyper-parameter controlling the number of Gaussian mixtures, $\mu_k$ and $\Sigma_k$ are the mean and covariance matrix of each Gaussian mixture. $\pi_k$, $\mu_k$ and $\log(\Sigma_k)$ are computed by the neural network layers before the GMM layer based on the input $s_t$ and $a_t$. Depending on our needs, the model can be learned by fitting state transitions and rewards either jointly or separately.

Once trained, the environment model can be used to examine problem formulation and data in many ways. One usage is to calculate feature importance and select only important features for RL training. We hypothesize that any feature with no importance in predicting state transitions or rewards should be discarded in order to reduce noise and increase learning efficiency. We use a heuristic that a feature's importance is the increase of the model loss due to masking the feature. The intuition is that if the feature is important, masking it would cause the model to perform much worse making the loss increase large. The current way to mask each feature is to set that feature to its mean value. Showing feature importance is also an effective way to help engineers examine datasets.

Another usage of the learned environment model is to evaluate problem formulation based on the definition of an MDP and heuristics. (1) We first check whether transitions are predictable by action and state features by looking at feature importance. An action or state feature is an important predictive feature if it increases the model loss when being masked, based on an environment model that fits only next states. An action suggested as not important means taking

the action would not exert influence on transitions, thus warranting further investigation on the design of the action space. On the other hand, if none of the state features are important in predicting next states, it indicates there is no sequential nature to the problem. (2) We check if there exists any state feature both dependent on actions and predictive of rewards. This verifies the reward is indeed determined by both actions and states in a meaningful way. When no state feature is predictive of rewards the problem would not pass the check: such problems can be reduced to multi-arm bandits where we just need to estimate the return of each action. The check also invalidates problems that pass the previous checks, but where no state feature involved in transitions is relevant to the rewards. We compute how dependent a state feature is to the actions taken by varying actions in the data and observing the extent to which the state features in the next state changes, based on the predictions of the environment model that only fits next states. We compute how predictive a state feature is of rewards by computing feature importance on a model fitting only rewards.

Although the data understanding tool is based on several heuristics that are not expected to cover all invalid problem formulations, in practice it has helped users understand the problem formulation in early stages of the RL training loop and has been effective at catching many improperly defined problems.

## 5. Model Implementations

Horizon contains implementations of several deep RL algorithms that span to solve discrete action, very large discrete action, and continuous action domains. We also provide default configuration files as part of Horizon so that end users can easily run these algorithms on our included test domains (e.g. OpenAI Gym (Brockman et al., 2016), Gridworld). Below we briefly describe the current algorithms supported in Horizon.

### 5.1. Discrete-Action Deep Q-Network (Discrete DQN)

For discrete action domains with a tractable number of actions, we provide a Deep Q-Network implementation (Mnih et al., 2015). We chose to include DQN in Horizon due to its relative simplicity and its importance as a building block for numerous algorithmic improvements (Hessel et al., 2017). In addition, we provide implementations for several DQN improvements, including double Q-learning (Van Hasselt et al., 2016), dueling architecture (Wang et al., 2015), and multi-step learning (Sutton et al., 1998). We plan on continuing to add more improvements to our DQN model (distributional DQN (Bellemare et al., 2017), and noisy nets (Fortunato et al., 2017)) as these improvements have been shown to stack to achieve state of the art results on numerous benchmarks (Hessel et al., 2017).

## 5.2. Parametric-Action Deep-Q Network (Parametric DQN)

Many domains at Facebook have have extremely large discrete action spaces (more than millions of possible actions) with actions that are often ephemeral. This is a common challenge when working on large scale recommender systems where an RL agent can take the action of recommending numerous different pieces of content. In this setting, running a traditional DQN would not be practical. One alternative is to combine policy gradients with a K-NN search (Dulac-Arnold et al., 2015), but when the number of available actions for any given state is sufficiently small, this approach is heavy-handed. Instead, we have chosen to create a variant of DQN called Parametric-Action DQN, in which we input concatenated state-action pairs and output the Q-value for each pair. Actions, along with states, are represented by a set of features. The rest of the system remains as a traditional DQN. Like our Discrete-Action DQN implementation, we also have adapted the double Q-learning and dueling architecture improvements to the Parametric-Action DQN.

## 5.3. Deep Deterministic Policy Gradients (DDPG) and Soft Actor-Critic (SAC)

Other domains at Facebook involve tuning of sets of hyperparameters. These domains can be addressed with a continuous action RL algorithm. For continuous action domains we have implemented Deep Deterministic Policy Gradients (DDPG) (Lillicrap et al., 2015) and Soft Actor-Critic (SAC) (Haarnoja et al., 2018). DDPG was selected for its simplicity and familiarity, while SAC was selected due to its recently demonstrated SOTA performance on numerous continuous action domains.

Support for other deep RL algorithms will be a continued focus going forward.

## 6. Training

Once we have preprocessed data and have a feature normalization function for each feature, we can begin training. Training can be done using CPUs, a GPU, or multiple GPUs across multiple machines. We utilize the PyTorch multi-GPU functionality to do distributed training (Paszke et al., 2017).

Using GPU and multi-GPU training we are able to train large RL models that contain hundreds to thousands of features across tens of millions of examples in a few hours (while doing feature normalization and counterfactual policy evaluation on every batch).

Typically, the initial RL policy is trained on off-policy data generated by a non-RL production policy. Once the first RL

policy is trained and deployed to a fraction of production traffic, subsequent training runs use this on-policy training data. In practice we have found that A/B test results improve as the RL model moves from learning on off-policy data to on-policy data. Figure 1 shows the change in the metric value of interest during a real A/B test.

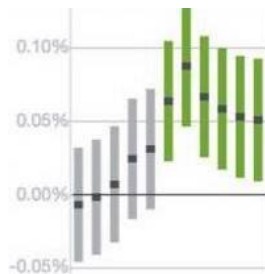

Figure 1. **Real RL model A/B Test Results.** The RL model (test) outperforms the non-RL model (control) on the push notification optimization task described in section 9.1. The x-axis shows the progression of the metric being optimized by day. Note, the performance of the RL model starts out neutral vs. the control, but quickly exceeds as it re-trains daily on data generated by itself.

Internally, we have recurring training jobs where models are updated on a daily frequency and training starts with the previous network weights and optimizer state (for stateful optimizers, e.g. Adam (Kingma & Ba, 2014)). Our empirical observations of performance improving as the RL policy learns from data generated by itself is inline with findings in literature. Specifically, recent literature has shown that off-policy RL aglorithims struggle significantly when learning from fixed batches of data generated under a seperate policy due to a phenomenon coined "extrapolation error" (Fujimoto et al., 2018). Extrapolation error is a phenomenon in which unseen state-action pairs are erroneously estimated to have unrealistic values. By retraining daily on self generated data, we mitigate this problem by forcing learning to be more "on-policy", thus improving the model performance.

## 7. Model Understanding And Evaluation

There are several features in Horizon that help engineers gain insight into each step of the RL model building loop (i.e. training, and evaluation). Below we describe the tools available at each step of the process:

- **Training:** Training metrics are surfaced that give insight into the stability and convergence of the training process.

- **Evaluation:** Several well known counterfactual policy evaluation estimates compute the expected performance of the newly trained RL model.

## 7.1. Training: TD-loss & MC-Loss

**Temporal difference loss (TD-loss)** measures the function approximation error. For example, in DQN, this measures the difference between the expected value of Q given by the bellman equation, and the actual value of Q output by the model. Note that, unlike supervised learning where the labels are from a stationary distribution, in RL the labels are themselves a function of the model and as a result this distribution shifts. As a result, this metric is primarily used to ensure that the optimization loop is stable. If the TD-loss is increasing in an unbounded way, we know that the optimization step is too aggressive (e.gs. the learning rate is too high, or the minibatch size is too small).

**Monte-Carlo Loss (MC-loss)** compares the model's Q-value to the logged value (the discounted sum of logged rewards). When the logged policy is the optimal policy (for example, in a toy environment), MC-loss is a very effective measure of the model's performance. Because the logged policy is often not the optimal policy, the MC-loss has limited usefulness for real-world domains. Similar to TD-loss, we primarily monitor MC-loss for extreme values or unbounded increase.

Because RL is focused on policy optimization, it is more useful to evaluate the policy (i.e. what action a model chooses) than to evaluate the model scores directly. Horizon has a comprehensive set of Counterfactual Policy Evaluation techniques.

## 7.2. Evaluation: Counterfactual Policy Evaluation

Counterfactual policy evaluation (CPE) is a set of methods used to predict the performance of a newly learned policy without having to deploy it online (Wang et al., 2017; Bottou et al., 2013; Dudık et al., 2011; Jiang & Li, 2016; Thomas & Brunskill, 2016). CPE is important in applied RL as deployed policies affect the real world. At Facebook, we serve billions of people every day; deploying a new policy directly impacts the experience they have using Facebook. Without CPE, industry users would need to launch numerous A/B tests to search for the optimal model and hyperparameters. These experiments can be time-consuming and costly. With reliable CPE, this search work can be fully automated using hyperparameter sweeping techniques that optimize for a model's CPE score. CPE also makes an efficient and principled parameter sweep possible by combining counterfactual offline estimates with real-world testing.

Horizon includes implementations of the following CPE estimators that are automatically run as part of training:

- Step-wise direct method estimator

- Step-wise importance sampling estimator (Horvitz & Thompson, 1952)

- Step-wise doubly-robust estimator (Dudık et al., 2011)

- Sequential doubly-robust estimator (Jiang & Li, 2016)

- Sequential weighted doubly-robust estimator (Thomas & Brunskill, 2016)

- MAGIC estimator (Thomas & Brunskill, 2016)

The first three estimators were originally designed to evaluate polices in contextual bandit problems (Auer et al., 2002; Langford & Zhang, 2008), the special cases of RL problems where the horizon of episodes is one. The step-wise direct method (DM) learns a reward function to estimate rewards that are not logged but expected to incur by the evaluated policy. The method suffers when the learned reward function has high bias. The step-wise importance sampling (IS) estimator (Horvitz & Thompson, 1952) uses action propensities of logged and evaluated policies to scale logged rewards in order to correct for different action distributions between the two policies. The step-wise IS estimator tends to have high variance (Dudık et al., 2011) and could be biased if logged action propensities are not accurate. The step-wise doubly-robust (DR) estimator (Dudık et al., 2011) combines the ideas of the previous two methods: (1) the bias tends to be low as long as either logged action propensities or the learned reward function is accurate; (2) the variance tends to be lower than the step-wise IS estimator under reasonable assumptions (Section 4 in (Dudık et al., 2011)). Due to these estimators' simplicity in the concept, we still compute them (averaging over steps) when evaluating longer episodes, though they will be biased.

The last three estimators are specifically designed for evaluating policies on longer horizons. The sequential DR estimator (Jiang & Li, 2016) inherits the advantage from the DR method that a low bias can be achieved if either action propensities or the reward function is accurate. The estimator has also been adapted to use weighted importance sampling (Thomas & Brunskill, 2016), which is considered to "better balance it (the bias-variance trade-off) while maintaining asymptotic consistency". In the same line of balancing the bias-variance trade-off, the MAGIC estimator (Thomas & Brunskill, 2016) combines the DR and DM in a way that directly optimizes the mean squared error (MSE).

Incorporating the aforementioned estimators into our platform's training pipeline provides us with two advantages: (1) all feature normalization improvements tailored to training are also available to CPE (2) users of our platform get CPE estimates at the end of each epoch which helps them understand how more training affects model performance. The CPE estimators in Horizon are also optimized for running speed. The implemented estimators incur minimal time overhead to the whole training pipeline.

One of the biggest technical challenges implementing CPE stems from the nature of how batch RL is trained. To decrease temporal correlation of the training data, which is needed for stable supervised learning, a pseudo i.i.d. environment is created by uniformly shuffling the collected training data (Mnih et al., 2015). However, the sequential doubly robust and MAGIC estimators both are built based on cumulative step-wise importance weights (Jiang & Li, 2016; Thomas & Brunskill, 2016), which require the training data to appear in its original sequence. In order to satisfy this requirement while still using the shuffled pseudo i.i.d. data in training, we sample and collect training samples during the training workflow. At the end of every epoch we then sort the collected samples to place them back in their original sequence and conduct CPE on the collected data. Such deferral provides the opportunity to calculate all needed Q-values together in one run, heavily utilizing matrix operations. As a side benefit, querying for Q-values at the end of one epoch of training decreases the variance of CPE estimates as the Q-function can be very unstable during training. Through this process we are able to calculate reliable CPE estimations efficiently. Internally, end users get plots similar to Figure 2 at the end of training. In the open source release we surface CPE results in TensorboardX.

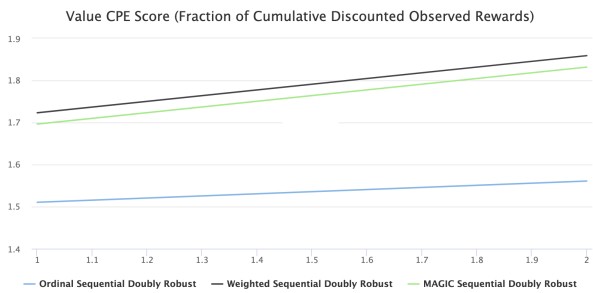

*Figure 2.* **Value CPE Results.** As part of training, Horizon surfaces CPE results indicating the expected performance of the newly trained policy relative to the policy that generated the training data. In this plot we see relative value estimates (y-axis) for several CPE methods vs. training time (x-axis) on a real Facebook dataset. A score of 1.0 means that the RL and the logged policy match in performance. These results show that the RL model should achieve roughly 1.5x - 1.8x as much cumulative reward as the logged system. As the number of training epochs increases, the CPE estimates improve.

### 7.3. TensorboardX

To visualize the output of our training process, we export our metrics to tensorboard using the TensorboardX plugin (Huang, 2018). TensorboardX outputs tensors from pytorch/numpy to the tensorboard format so that they can be viewed with the Tensorboard web visualization tool.

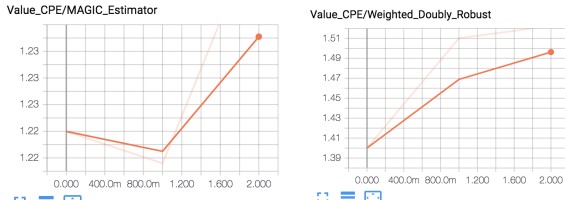

*Figure 3.* **TensorboardX CPE Results.** Example TensorboardX counterfactual policy evaluation results on the CartPole-v0 environment. The x-axis of each plot shows the number of epochs of training and the y-axis shows the CPE estimate. While we only display two CPE methods here (MAGIC and Weighted Doubly Robust), several other CPE methods and loss plots are displayed in the final Tensorboard dashboard post-training. In these plots a score of 1.0 means that the RL and the logged policy match in performance. Here we see the RL model should achieve roughly 1.2x - 1.5x as much cumulative reward as the logged policy.

## 8. Model Serving

At Facebook, we serve deep reinforcement learning models in a variety of production applications.

PyTorch 1.0 supports ONNX (Exchange, 2018), an open source format for model inference. ONNX works by tracing the forward pass of an RL model, including the feature transformation and the policy outputs. The result is a Caffe2 network and a set of parameters that are serializable, portable, and efficient. This package is then deployed to thousands of machines.

At serving time, product teams run our RL models and log the possible actions, the propensity of choosing each of these actions, the action chosen, and the reward received. Depending on the problem domain, it may be hours or even days before we know the reward for a particular sample. Product teams typically log a unique key with each sample so they can later join the logged training data to other data sources that contain the reward. This joined data is then fed back into Horizon to incrementally update the model. Although all of our algorithms are off-policy, they are still limited based on the policy that they are observing, so it is important to train in a closed loop to get the best results. In addition, the data distribution is changing and the model needs to adapt to these changes over time.

## 9. Real World Deployment: Notifications at Facebook

### 9.1. Push Notifications

Facebook sends notifications to people to connect them with the most important updates when they matter, which may include interactions on your posts or stories, updates about your friends, joined groups, followed pages, interested

events etc. Push notifications are sent to mobile devices and a broader set of notifications is accessible from within the app/website. It is primarily used as a channel for sending personalized and time sensitive updates. To make sure we only send the most personally relevant notifications to people, we filter notification candidates using machine learning models. Historically, we have used supervised learning models for predicting click through rate (CTR) and likelihood that the notification leads to meaningful interactions. These predictions are then combined into a score that is used to filter the notifications. For example, this score could look like:

$$score = weight_1 * P(event_1) + weight_2 * P(event_2) + ...$$

This however, didn't capture the long term or incremental value of sending notifications. There can be some signals that appear long after the decision to send or drop is made or that can't be attributed directly to the notification.

We introduced a new policy that uses Horizon to train a Discrete-Action DQN model for sending push notifications to address the problems above. The Markov Decision Process (MDP) is based on a sequence of notification candidates for a particular person. The actions here are sending and dropping the notification, and the state describes a set of features about the person and the notification candidate. There are rewards for interactions and activity on Facebook, with a penalty for sending the notification to control the volume of notifications sent. The policy optimizes for the long term value and is able to capture incremental effects of sending the notification by comparing the Q-values of the send and drop action. Specifically, the difference in Q-values is computed and passed into a sigmoid function to create an RL based policy:

$$\begin{cases} send; & \text{if } sigmoid(Q(send) - Q(drop)) \geq threshold \\ drop; & \text{if } sigmoid(Q(send) - Q(drop)) < threshold \end{cases}$$

If the difference between $Q(send)$ and $Q(drop)$ is large, this means there is significant value in sending the notification. If this difference is small, it means that sending a notification is not much better than not sending a notification.

As an implementation trick, we use a proportional integral derivative (PID) controller to tune the threshold used in the RL policy. This helps to keep the RL policy's action distribution inline with the previous production policy's action distribution.

The model was incrementally retrained daily on data from people exposed to the model with some action exploration introduced during serving. The model is updated with batches of tens of millions of state transitions. We observed

this to help online usage metrics as we are doing off-policy batch learning. The benefit of this is shown in figure 1.

We observed a significant improvement in activity and meaningful interactions by deploying an RL based policy for certain types of notifications, replacing the previous system based on supervised learning.

### 9.2. Page Administrator Notifications

In addition to Facebook users, page administrators also rely on Facebook to provide them with timely updates about the pages they manage. In the past, supervised learning models were used to predict how likely page admins were to be interested in such notifications and how likely they were to respond to them. Although the models were able to help boost page admins' activity in the system, the improvement always came at some trade-off with the notification quality, e.g. the notification click through rate (CTR). With Horizon, a Discrete-Action DQN model is trained to learn a policy to determine whether to send or not send a notification based on the state represented by hundreds of features. The training data spans multiple weeks to enable the RL model to capture page admins' responses and interactions to the notifications with their managed pages over a long term horizon. The accumulated discounted rewards collected in the training allow the model to identify page admins with long term intent to stay active with the help of being notified. After deploying the DQN model, we were able to improve daily, weekly, and monthly metrics without sacrificing notification quality.

### 9.3. More Applications of Horizon

In addition to making notifications more relevant on our platform, Horizon is applied by a variety of other teams at Facebook. The 360-degree video team has applied Horizon in the adaptive bitrate (ABR) domain to reduce bitrate consumption without harming people's watching experience. This was due to more intelligent video buffering and pre-fetching.

While we focused our case studies on notifications, Horizon is a horizontal effort in use or being explored to be used by many organizations within Facebook.

## 10. Future Work

The most immediate future additions to Horizon will be new models & model improvements. We will be adding more incremental improvements to our current models and plan on continually adding the best performing algorithms from the research community.

We welcome community pull requests, suggestions, and feedback.

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
