# OpenReview forum: "Horizon: Facebook’s Open Source Applied Reinforcement Learning Platform"
_ICML.cc/2019/Workshop/RL4RealLife — RL4RealLife 2019_

### Official Review · AnonReviewer2 · 2019-05-25
**Detailed description of a State-of-the-art RL Platform, with Real-world applications at FB**

**Rating:** 4
**Confidence:** 5

**Review:**

This paper introduces the Horizon RL platform developed at Facebook. Not only do the authors focus on the RL algorithms, they also spend significant efforts explaining the data collection and processing pipeline, off-policy evaluation criteria etc., which are fundamentally important in RL for real-life applications. Furthermore, the paper also dives deep into the notification example at FB where a Q-learning approach is deployed to recommend push notifications.

The main contribution of this paper comes from the detailed descriptions on the implementation of the Horizon RL platform,  ranging from data, algorithm and evaluations. I also appreciate the effort of open-sourcing codes associated with the paper. Furthermore, the real-life examples discussed in this paper is one of the few examples of using RL in industrial scale, which is impressive.

In general I think this paper is above threshold and I would recommend acceptance of this paper. The only (minor) question/comment is about reproducibility of the FB results. But I understand there might be challenges/difficulties in open-sourcing proprietary data (even anonymous) to the community to test various RL algorithms.

---

### Official Review · AnonReviewer1 · 2019-05-25
**Good paper summarizing the facebook applied reinforcement learning platform**

**Rating:** 5
**Confidence:** 4

**Review:**

The paper presents Horizon, facebook's open source platform for tackling industrial-scale applied RL problems. Compared with other open sourced RL platforms, Horizon offers data preprocessing pipeline, and counterfactual policy evaluation, as well as some data and model understanding functionality. The implemented algorithms are more targeted toward industry RL problems with large state and action spaces. The authors showcase the effectiveness of the platform by deploying it in several Facebook products.

The paper is well written and pleasant to read. Here're a few points to improve the platform:
1. Off-policy learning is prevalent in industry. The policy-based algorithms supported by the platform currently only handles on-policy learning.  It might worth to include a couple of off-policy algorithms;
2. Add confidence intervals for counterfactual policy estimators;
3. Support bandits use case.

The exact formulation used in the push notification is not clear from the writing. Is it using SARSA?

---

### Decision · Program_Chairs · 2019-05-28

Accept